# Diversity and structure of prokaryotic communities within organic and conventional farming systems in central highlands of Kenya

Edward Nderitu Karanja[1,2], Andreas Fliessbach[3], Noah Adamtey[3], Anne Kelly Kambura[4], Martha Musyoka[2], Komi Fiaboe[2,5], Romano Mwirichia[1]*

**1** Department of Biological sciences, University of Embu, Embu, Kenya, **2** International Centre for Insect Physiology and Ecology, Nairobi, Kenya, **3** Research Institute of Organic Agriculture, Frick, Switzerland, **4** Taita Taveta University, School of Agriculture, Earth and Environmental Sciences, Voi, Kenya, **5** International Institute of Tropical Agriculture, Cameroon, Yaoundé, Cameroon

* mwirichia.romano@embuni.ac.ke

**Data Availability Statement:** Sequence data is available on NCBI Sequence Read Archive with SRA accession: PRJNA523239 (https://www.ncbi.

## Abstract

Management practices such as tillage, crop rotation, irrigation, organic and inorganic inputs application are known to influence diversity and function of soil microbial populations. In this study, we investigated the effect of conventional versus organic farming systems at low and high input levels on structure and diversity of prokaryotic microbial communities. Soil samples were collected from the ongoing long-term farming system comparison trials established in 2007 at Chuka and Thika in Kenya. Physicochemical parameters for each sample were analyzed. Total DNA and RNA amplicons of variable region (V4—V7) of the 16S rRNA gene were generated on an Illumina platform using the manufacturer's instructions. Diversity indices and statistical analysis were done using QIIME2 and R packages, respectively. A total of 29,778,886 high quality reads were obtained and assigned to 16,176 OTUs at 97% genetic distance across both 16S rDNA and 16S rRNA cDNA datasets. The results pointed out a histrionic difference in OTUs based on 16S rDNA and 16S rRNA cDNA. Precisely, while 16S rDNA clustered by site, 16S rRNA cDNA clustered by farming systems. In both sites and systems, dominant phylotypes were affiliated to phylum *Actinobacteria*, *Proteobacteria* and *Acidobacteria*. Conventional farming systems showed a higher species richness and diversity compared to organic farming systems, whilst 16S rRNA cDNA datasets were similar. Physiochemical factors were associated differently depending on rRNA and rDNA. Soil pH, electrical conductivity, organic carbon, nitrogen, potassium, aluminium, zinc, iron, boron and micro-aggregates showed a significant influence on the observed microbial diversity. The observed higher species diversity in the conventional farming systems can be attributed to the integration of synthetic and organic agricultural inputs. These results show that the type of inputs used in a farming system not only affect the soil chemistry but also the microbial population dynamics and eventually the functional roles of these microbes.

nlm.nih.gov/sra/PRJNA523239) and SRA accession: PRJNA523223 (https://www.ncbi.nlm. nih.gov/Traces/study/?acc=PRJNA523223) for 16S rDNA and 16S rRNA cDNA datasets, respectively.

**Funding:** This research was financially supported by Biovision Foundation, Swiss Coop Sustainability Fund, Liechtenstein Development Service and Swiss Agency for Development and Cooperation through Research Institute of Organic Agriculture, the kind contribution by International Centre for Insect Physiology and Ecology core funding provided by UK-Aid from UK Government, Swedish International Development Cooperation Agency, Swiss Agency for Development and Cooperation, Federal Democratic Republic of Ethiopia and Kenyan Government.The funders had no role in study design, data collection and analysis, decision to publish, or preparation of the manuscript.

**Competing interests:** The authors have declared that no competing interests exist.

## Introduction

Microorganisms play an important role in soil fertility by carrying out biochemical transformations thereby making soil a source and sink of mineral nutrients. Plant-associated microbes colonize both exterior and interior plant surfaces, while surrounding soil acts as major source for resources needed by microbes [1]. There is usually a dynamic interaction between plant and microorganisms in different agricultural ecosystems [2]. Agricultural management effects on the soil microbial communities are complex and diverse [3, 4] and retrieving comprehensively effective explanations on organic and conventional farming systems is thought-provoking. Management practices influence soil microbial community structure [5] hence, intensive farming practices may undermine the welfare of natural habitats leading to disruption of ecosystem services [6]. Although it has been suggested that low-input farming systems promote higher abundance and diversity of most organisms [7], studies conducted in the last ten (10) years have not conclusively established the beneficial effects or otherwise of organic agriculture on microbial diversity and plant-associated microorganisms [8, 9, 10, 11]. Therefore, understanding how changes in land management affect soil microbial community structure could provide an important index for assessing the relative ability of soils to respond to future disturbance [12, 13]. Long-term experiments on farming systems, especially when compared to medium and/or short-term experiments can generate important information to predict the dynamics of the soil microbial community with time. High throughput sequencing of both DNA and RNA has proven to be a powerful tool that provides valuable insights about the structure, functions, and interactions of different microbial communities in an ecosystem [14, 15]. These methods involve direct isolation and analysis of nucleic acids from samples [14, 15, 16, 17, 18] and assist in exploration of mixed microbial communities existing in various natural environments [19, 20]. In this study, we used amplicon sequencing of the 16S rDNA and 16S rRNA cDNA genes to create a taxonomic profile of soil prokaryotic communities in long-term experiment study sites located at Chuka and Thika within central highlands of Kenya.

## Materials and methods

### Study sites characteristics

The study was done in the ongoing long term experiment trial sites established in 2007 [21] at Chuka and Thika in the sub-humid zones of central highlands in Kenya (https://systems-comparison.fibl.org/). The study sites were initiated by the Research Institute of Organic Agriculture (FiBL) and their local partners; International Centre for Insect Physiology and Ecology (*icipe*) and Kenyan Agricultural and Livestock Research Organization (KALRO) to compare productivity, profitability and sustainability of organic and conventional farming systems in the tropics. These sites were established based on Food and Agricultural Organization (FAO) world reference system of soil classification. The soil at Thika site is classified as Rhodic Nitisol, while that of Chuka is classified as Humic Nitisol [22]. The site characteristics are as summarized in Table 1.

**Table 1. Long term SysCom experiment trial sites characteristics.**

| Site | Coordinates | Agro ecological Zone | Altitude | Rainfall pattern | Temperature Range | Cropping Seasons | Cropping Period |
|------|-------------|----------------------|----------|------------------|-------------------|------------------|-----------------|
| Thika | 01˚ 0.231' S 37˚ 04.747' E | UM 3 | 1518 m | 840 mm | 19.5–20.7˚C | Long Rain | March—June |
|       |             |                      |          |                  |                   | Short Rain | October—December |
| Chuka | 0˚ 20.864' S 37˚ 38.792' E | UM 2 | 1458 m | 1373 mm | 19.2–20.6˚C | Long Rain | March—June |
|       |             |                      |          |                  |                   | Short Rain | October—December |

UM 2 –Main Coffee Zone b) UM 3 –Sunflower and Maize Zone.

## Farming systems

Conventional (Conv) and organic (Org) farming systems were compared at low input levels (Conv-Low and Org-Low), where nitrogen and phosphorous application rates mirrored small-scale farmers' practices in the region; and at high input levels (Conv-High and Org-High), which represented the recommended nitrogen and phosphorous input levels used in market-oriented and large-scale production systems. In Conv-High system, nutrients were applied in form of synthetic fertilizers (diammonium phosphate, triple super phosphate, calcium ammonium nitrate) and decomposed manure. Nutrient application rate was based on recommendations by [23], while in Org-High system, nutrients were applied in form of compost, green manure, plant tea and phosphate rock [24] at the same nutrient levels for Phosphorus and Nitrogen as in Conv-High system. The high input systems received supplementary irrigation during the dry period and pest and disease were controlled based on a scouting program [21]. In the low input conventional and organic farming systems, nutrients were applied in form of synthetic fertilizers and fresh farmyard manure (Conv-Low) and decomposed manure, biomass of *Tithonia diversifolia* and low amounts of phosphate rock (Org-Low) (S1 Table).

In both sites, the four farming systems were randomly replicated four times. At Chuka, the replicates were designated as; Conv-High (plots 3C, 6C, 12C and 14C), Conv-Low (plots 2C, 7C, 11C and 16C), Org-High (plots 4C, 8C, 9C and 15C) and Org-Low (plots 1C, 5C, 10C and 13C). At Thika, the replicates were designated as; Conv-High (plots T2, T7, T9 and T20), Conv-Low (plots T1, T6, T12 and T18), Org-High (plots T3, T8, T11 and T17) and Org-Low (plots T4, T5, T10 and T19) (Fig 1A and 1B).

## Soil sampling and physicochemical analysis

Soil sampling was done before land preparation in March 2015. Surface organic materials were removed and a composite soil sample collected from 12 single cores within top soil (0–20 cm depth) which is the root zone of majority crops grown in the trial sites. Two batches of sixteen (16) composite samples from each site were packed in sterile 500 g containers. Samples for molecular analysis were preserved on dry ice and transported to the laboratory at *icipe* for

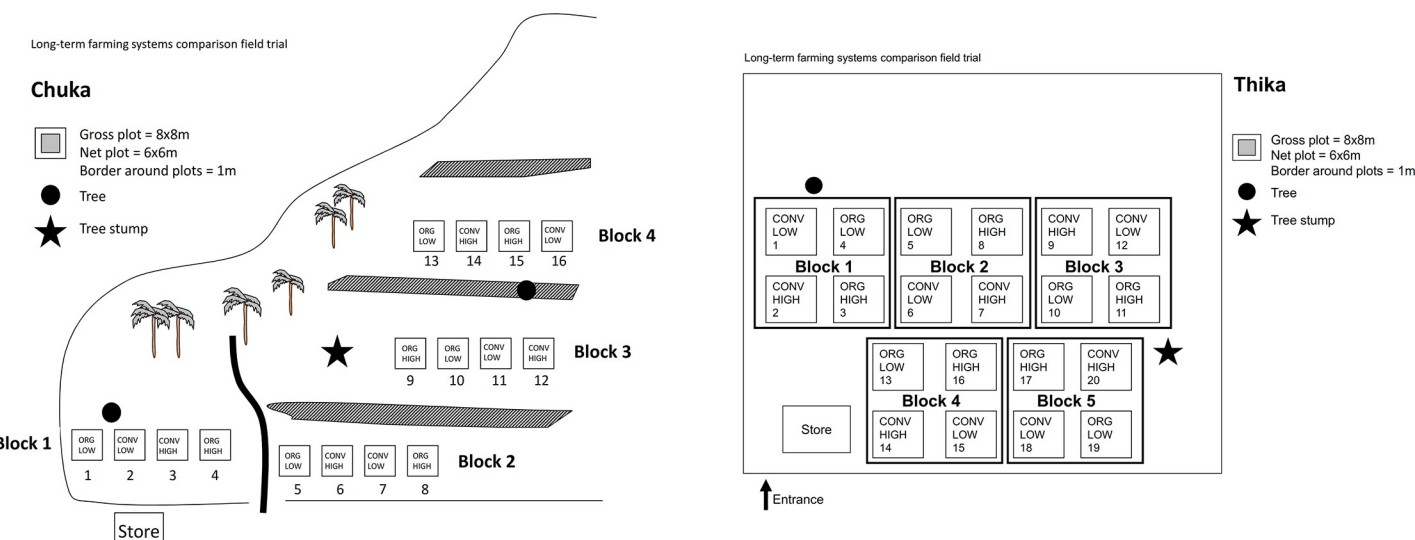

**Fig 1.** a. Chuka long-term farming system comparison experiment field trial layout. b. Thika long-term farming system comparison experiment field trial layout.

**Table 2. Soil physicochemical parameters analyzed and their respective methods.**

| Parameter | Method |
|---|---|
| pH and Electrical conductivity (EC) | Potentiometric [25] |
| Cation exchange capacity (CEC), Potassium (K), Calcium (Ca), Magnesium (Mg), Sulphur (S), Sodium (Na), Copper (Cu), Boron (B), Zinc (Zn) and Iron (Fe) | Mehlich 3 [26] |
| Exchangeable Aluminium (Exch. Al) | Spectrophotometry [27] |
| Organic Carbon (OC) | Wet oxidation [28] |
| Total Nitrogen (N) | Kjeldahl acid digestion [29] |
| Total Phosphorous (P), | Olsen [25] |
| Soil moisture and Temperature | Soil Moisture Meter (IMKO GmbH–Germany) |
| Aggregate size separation (Small macro-aggregates and micro-aggregates) | Wet sieving [30] |
| Soil mineralogy | Diffraction [31] |

preservation at -80 ℃ whilst the batch of samples for physicochemical analysis were transported to the laboratory at *icipe* and preserved at room temperature. Soil physicochemical parameters were analyzed using methods summarized in Table 2.

## Microbial community analysis

Total community DNA was extracted from 0.2 g of the soil samples in triplicates exactly as described [32]. Total RNA was extracted from 0.25 g of soil samples in triplicates using Trizol RNA extraction protocol [33]. The respective nucleic acids extracted from triplicate samples were pooled during the precipitation stage, pellets air dried and sent to Molecular Research DNA Lab (www.mrdnalab.com, Shallowater, TX, USA) for cDNA synthesis [34], amplicon generation and sequencing. PCR amplification of the 16S rRNA gene V4 variable region was carried out from extracted DNA and cDNA generated from rRNA, using barcoded bacteria/archaeal primers 515F (5′-GTGCCAGCMGCCGCGGTAA-3′) and 806R (5′-GGACTACHV GGGTWTCTAAT-3′) as described [35]. Sequencing was performed on a MiSeq 2x300bp Version 3 following the manufacturer's guidelines.

## Bioinformatic sequence processing and taxonomic identification

The generated amplicons were analyzed using QIIME2 pipeline [36]. The FASTQ sequences were demultiplexed, quality checked and a feature table constructed using dada2 [37]. This pipeline denoises sequences, removes chimeras, creates OTU table, picks representative sequences and calculates denoising statistics. Sequences which were < 200 base pairs after phred20- base quality trimming, with ambiguous base calls, and those with homopolymer runs exceeding 6bp were removed. Representative sequences were aligned using MAFFT and highly variable regions were masked to reduce the noise in phylogenetic analysis [38]. Phylogenetic trees for use in phyloseq analysis were created and rooted at midpoint [39]. Taxonomic classification of representative sequences obtained from the OTU clustering was done using QIIME feature-classifier [36]. Sequences were submitted to NCBI Sequence Read Archive with SRA accession: PRJNA523239 (https://www.ncbi.nlm.nih.gov/sra/PRJNA523239) and SRA accession: PRJNA523223 (https://www.ncbi.nlm.nih.gov/Traces/study/?acc=PRJNA523223) for 16S rDNA and 16S rRNA cDNA datasets, respectively. Microbial diversity analysis was carried out using Vegan Community Ecology Package version 2.5.2 [40] while microbiome census was analyzed using phyloseq version 1.24.2 in R [41] (R Development Core Team, 2012). Alpha diversity measures (Richness—S' and Shannon—H') were used to test significant

differences between high and low input farming systems. Rarefaction curves were generated, plotted and customized using Vegan Community Ecology Package [40]. Community and environmental distances were compared using Analysis of similarity (ANOSIM) while significance was determined at 95% confidence interval (P<0.05). Calculation of Bray-Curtis dissimilarities between datasets and hierarchical clustering were carried out using Vegan package in R [40]. Diversity between samples (β diversity) was estimated by computing the Principal Component Analysis (PCA) of soil physicochemical characteristics versus prokaryotic taxa in R [41]. In order to understand the influence of farming systems on soil physicochemical characteristics, analysis of variance was performed at P < 0.05, 0.01 and 0.001 using a linear mixed-effect model with *lmer* function from lme4 package [42] with system and site as fixed effects, while replication was used as random effect. Computation of least mean squares was done using *lsmeans* package. Means were separated using Tukey's *ad hoc* method implemented using *cld* from *multicomp* package as developed by [43] in R software version Ri386 3.1.1 [44].

## Results

### General sequence analysis

After demultiplexing, quality filtering, denoising, and removal of potential chimeras, 476,103 and 632,573 high quality sequences were obtained from 16S rDNA and 16S rRNA cDNA datasets, respectively at Chuka site. These were clustered into 4,916 and 530 OTUs at 97% genetic distance in 16S rDNA and 16S rRNA cDNA datasets, respectively. The 16S rDNA OTUs were further classified into 29 phyla, 96 classes and 166 orders while 16S rRNA cDNA OTUs were assigned to 14 phyla, 30 classes and 52 orders. At Thika site, 931, 400 and 937,810 high quality sequences were obtained from 16S rDNA and 16S rRNA cDNA datasets, respectively. These were clustered into 10,082 and 648 OTUs at 97% genetic divergence in 16S rDNA and 16S rRNA cDNA datasets, respectively. The 16S rDNA OTUs were assigned to 35 phyla, 123 classes and 229 orders while 16S rRNA cDNA OTUs were assigned to 14 phyla, 35 classes and 57 orders within prokaryotic domain (Table 3). Composition and diversity assessment of prokaryotic communities within sites and farming systems displayed Thika site to harbor more unique OTUs as compared to Chuka site. For instance, at Thika site, Conv-High (2,444) and Org-Low (1,633) systems had the highest number of unique OTUs within 16S rDNA dataset.

Bacterial groups were the most abundant within datasets at both sites. The top 10 most abundant classes of bacteria comprised *Alphaproteobacteria*, *Actinobacteria*, *Thermoleophila*, *Unknown phyla*, *Bacillus*, *Blastocatellia*, *Betaproteobacteria*, *Acidimicrobia*, *Solibacteres* and *Gammaproteobacteria*. Archaeal groups were represented by *Thaumarchaeota* and *Euryarchaeota*. The distribution of high-quality sequences, OTUs and prokaryotic taxa are summarized in Table 3; while the most predominant phyla within each dataset are as shown on Fig 2.

Comparison of prokaryotic diversity at order level within 16S rDNA, revealed 79 and 115 shared orders across all farming systems at Chuka and Thika sites respectively. The number of unique taxa within each farming system are indicated in (Fig 3A–3F) at Chuka site and (Fig 3G–3L) at Thika site. Twenty one (21) and 35 prokaryotic orders were shared across all farming systems at Chuka (Fig 4A and 4B) and Thika (Fig 4G and 4H) sites respectively, within 16S rRNA cDNA dataset. Unique taxa within 16S rRNA cDNA dataset are shown in (Fig 4C–4F) at Chuka and (Fig 4I–4K) at Thika sites. Mean abundances of the most notable bacterial and archaeal orders in each farming system indicated *Proteobacteria* orders (*Caulobacterales*, *Rhizobiales*, *Burkholderiales*, *Sphingomonadales*, *Pseudomonadales* and *Enterobacteriales*); *Actinobacteria* orders (*Acidimicrobiales*, *Corynebacteriales*, *Solirubrobacterales* and *Gaiellales*); and *Firmicutes* (*Bacillales* and *Lactobacillales)* as key drivers of biological processes. The mean abundances are summarized on (S2 and S3 Tables).

**Table 3. Distribution of high-quality sequences, OTUs, diversity indices and prokaryotic taxa at Chuka and Thika sites sorted as per total number of OTUs.**

| | Site | System | High quality sequences | OTUs | Unique OTUs | Richness | Shannon (H) | Phyla | Classes | Orders | Unknown orders | Most abundant taxa (order level) |
|---|---|---|---|---|---|---|---|---|---|---|---|---|
| **16S rDNA** | Thika | Conv-High | 319678 | 3193 | 2444 | 877.2 | 6.26 | 19 | 97 | 170 | 81 | *Solirubrobacterales* |
| | | Org-High | 182931 | 2314 | 1565 | 757.5 | 6.09 | 27 | 87 | 151 | 68 | *Uncultured Chloroflexi* |
| | | Org-Low | 207067 | 2307 | 1633 | 823.4 | 6.12 | 29 | 87 | 144 | 62 | *Burkholderiales* |
| | | Conv-Low | 221724 | 2268 | 1594 | 728.6 | 6.09 | 27 | 83 | 154 | 66 | *Uncultured Chloroflexi* |
| | Chuka | Conv-Low | 108652 | 1737 | 1400 | 407 | 5.29 | 23 | 77 | 120 | 45 | *Gaiellales* |
| | | Conv-High | 115842 | 1497 | 1210 | 358 | 4.74 | 21 | 64 | 110 | 36 | *Sphingomonadales* |
| | | Org-Low | 145520 | 862 | 525 | 405.5 | 5.33 | 23 | 72 | 119 | 46 | *Acidimicrobiales* |
| | | Org-High | 106089 | 820 | 533 | 350.25 | 5.08 | 23 | 71 | 111 | 41 | *Acidimicrobiales* |
| **16S rRNA cDNA** | Thika | Org-High | 230728 | 174 | 75 | 81 | 2.56 | 12 | 25 | 41 | 7 | *Corynebacteriales* |
| | | Conv-High | 242725 | 164 | 65 | 72.6 | 1.68 | 13 | 29 | 49 | 12 | *Rhizobiales* |
| | | Conv-Low | 181506 | 160 | 73 | 76 | 2.66 | 12 | 24 | 43 | 11 | *Corynebacteriales* |
| | | Org-Low | 282851 | 150 | 63 | 65 | 1.77 | 12 | 26 | 42 | 9 | *Corynebacteriales* |
| | Chuka | Conv-Low | 156088 | 144 | 67 | 62 | 2.4 | 11 | 23 | 40 | 7 | *Enterobacteriales* |
| | | Org-Low | 193582 | 136 | 59 | 58 | 1.55 | 11 | 22 | 37 | 6 | *Rhizobiales* |
| | | Org-High | 122091 | 126 | 63 | 54.75 | 2.05 | 11 | 22 | 37 | 4 | *Rhizobiales* |
| | | Conv-High | 160812 | 124 | 61 | 55.75 | 2.03 | 11 | 19 | 35 | 6 | *Rhizobiales* |

## Diversity indices of soil prokaryotic communities

Alpha diversity indices within farming systems and sites showed no significant difference (P>0.05) in Richness (S) and Shannon index (H'). However, at Thika there was a higher species richness and the communities were more diverse (H) compared to Chuka (Table 3). At Chuka site, low input farming systems were found to exhibit higher total species richness (Conv-Low = 407.00 and Org-Low = 405.50) as compared to high farming systems (Conv-High = 358.00 and Org-High = 350.25). At Thika, Conv-High had higher total species richness (877.2) and diversity (H = 6.26) but Org-High and Conv-Low exhibited higher active species richness (81) and active species diversity (H = 2.66), respectively. Analysis of similarity pointed to highly significant differences between OTUs within high and low input farming systems (P<0.001) at Chuka site. However, there were no significant differences observed at Thika site (ANOSIM P<0.672 and 0.241 within 16S rDNA and 16S rRNA cDNA datasets, respectively). The prokaryotic taxa in each farming system were visualized using rarefaction curves. A steep slope that flattened to the right was observed in the rarefaction curves indicating that a reasonable number of prokaryotic groups had been sequenced and more intensive sampling was likely to yield only a few additional species. The sampling curves tended to be asymptotic, denoting that prokaryotic communities were relatively deeply sampled (Fig 5A–5D).

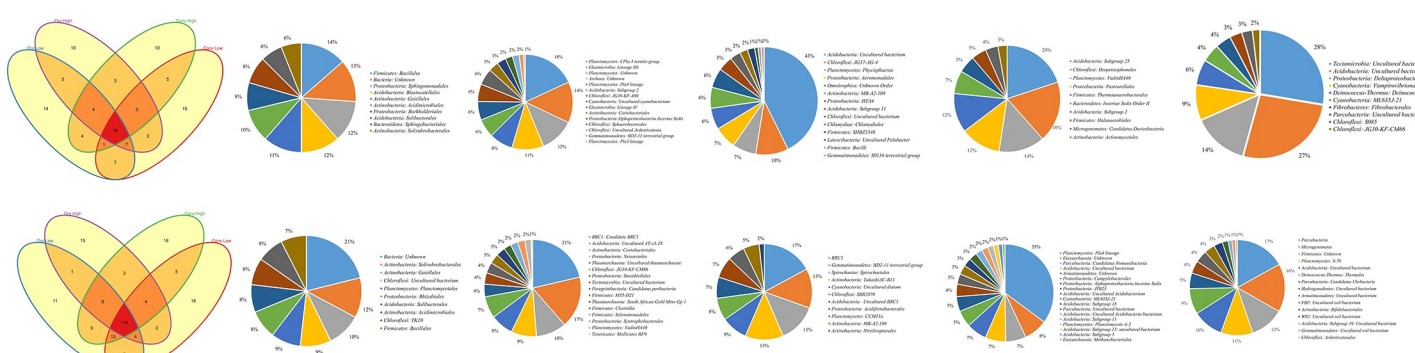

**Fig 2. Relative abundance of the most predominant phyla in both datasets at Chuka and Thika sites.**

**Fig 3. a-f.** Shared and unique prokaryotic taxa in 16S rDNA at Chuka. The Venn diagram (3a) show number of shared and unique taxa at order level within farming systems. The pie diagrams (3b - f) show most abundant and unique taxa at order level across farming systems. **g-l.** Shared and unique prokaryotic taxa in 16S rDNA at Thika. The Venn diagram (3g) show number of shared and unique taxa at order level within farming systems. The pie diagrams (3h - l) show most abundant and unique taxa at order level across farming systems.

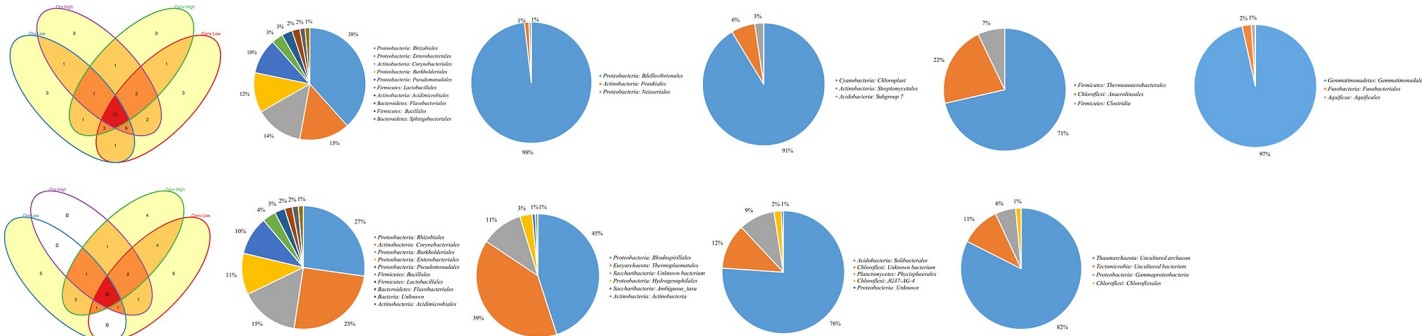

**Fig 4.  a-f.** Shared and unique prokaryotic taxa in 16S rRNA cDNA at Chuka. The Venn diagram (4a) show number of shared and unique taxa at order level within farming systems. The pie diagrams (4b - 4f) show most abundant and unique taxa at order level across farming systems. **g-k.** Shared and unique prokaryotic taxa in 16S rRNA cDNA at Chuka. The Venn diagram (4g) show number of shared and unique taxa at order level within farming systems. The pie diagrams (4h - k) show most abundant and unique taxa at order level across farming systems.

## Soil physicochemical properties for the different sites

In this study we assessed the prokaryotic community composition in 32 soil samples collected from long-term farming system comparison trials at Chuka and Thika in Kenya. The physico-chemical characteristics for the samples analysed are presented (Table 2). Tukey's separation of means revealed a trend in the means of soil pH, P, K, Ca, Mg, B and small macro-aggregates that were found to be significantly higher (P<0.05) in Org-High farming system. Higher means of Fe and micro-aggregates were recorded in Conv-High and Conv-Low systems, respectively (Table 4). Soils from Chuka contained as much as 59.4% primary clay minerals and 40.6% secondary clay minerals, while soils from Thika were characterized by high primary minerals (78.3%) and low secondary clay minerals (21.7%). Congruently, the rate of formation and stabilization of macro aggregates was found to be higher at Thika than Chuka site.

## Key environmental drivers of prokaryotic communities

In order to assess how environmental variables shaped soil prokaryotic community structure, PCA was performed on soil physicochemical characteristics within farming systems and pro-karyotic taxa at species level. Each characteristic was assessed on its ability to influence diversity positively or negatively within sites and farming systems. At Chuka, pH, OC, N, Zn, Fe and Al were found to be the major drivers of prokaryotic diversity within farming systems while at Thika, key properties displayed were pH, EC, OC, N, K, Fe, Zn, B and micro-aggregate (MA) as shown on Fig 6A–6D.

The relationship between most predominant phyla within both datasets in the two study sites and farming systems was analyzed using hierarchical clustering. Heatmaps revealed clustering of sites into two major groups while farming systems clustered into four sets on the dendogram, representing the two sites, each with four farming systems under investigation. There was an indication that farming systems in both sites harbored prokaryotic taxa within active diversity dataset which possibly interacted with one another to perform essential ecological functions as shown on Fig 7A and 7B.

## Discussion

In this study, we used experimentally manipulated farming systems and high-throughput sequencing of 16S rDNA and 16S rRNA cDNA amplicons to demonstrate that farming inputs whether organic or conventional have an immense influence on the prokaryotic community

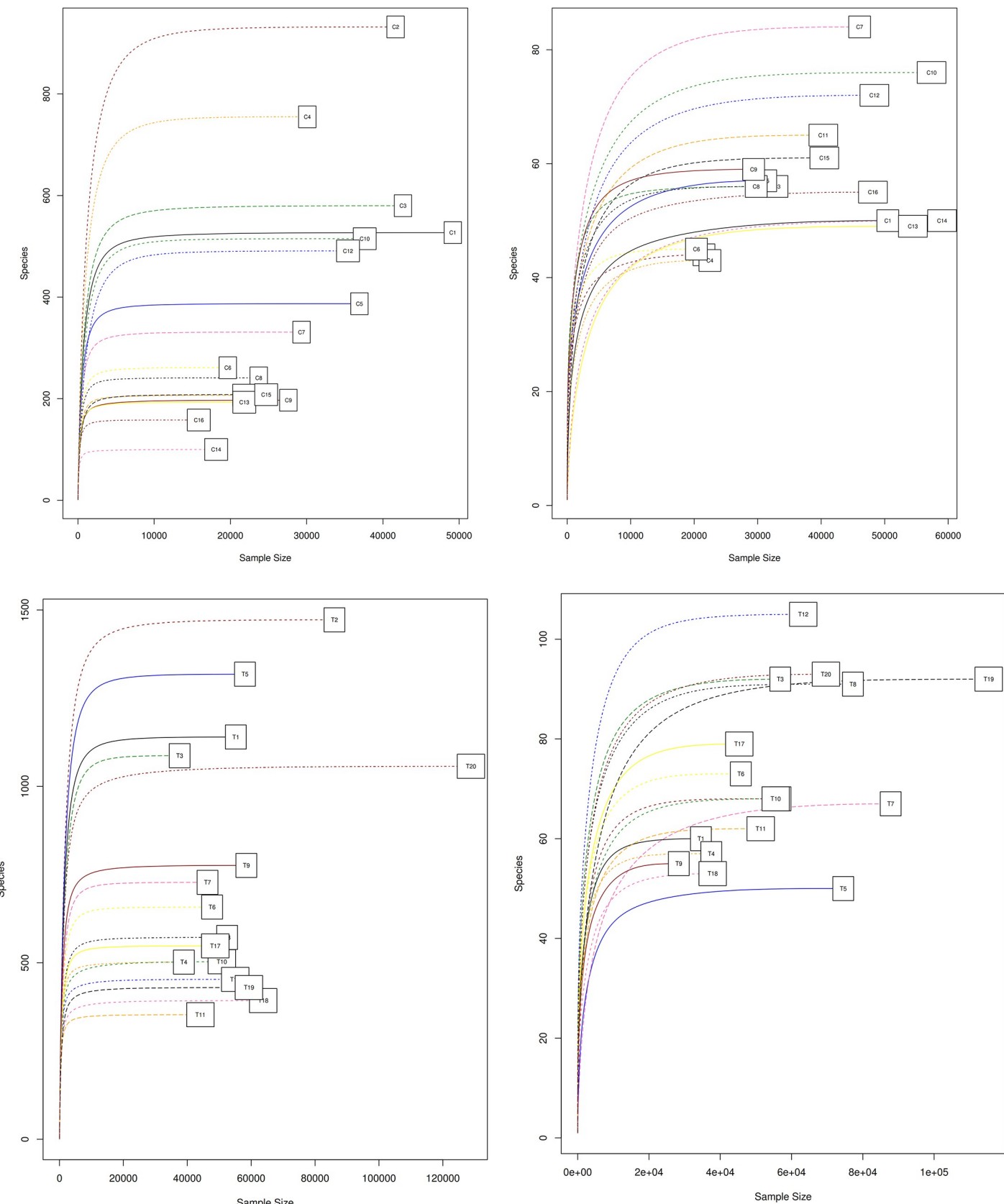

**Fig 5. a-d.** Rarefaction curves indicating level of sequence coverage.

**Table 4. Soil physicochemical characteristics as influenced by farming systems.**

| | Farming Systems | | | | System x Site | | | | | | | | Source of variation | |
| | | | | | Chuka | | | | Thika | | | | System | System x Site |
| | Conv-High | Org-High | Conv-Low | Org-Low | Conv-High | Org-High | Conv-Low | Org-Low | Conv-High | Org-High | Conv-Low | Org-Low | | |
|---|---|---|---|---|---|---|---|---|---|---|---|---|---|---|
| **pH** | 5.68[a] | 6.61[ab] | 5.43[a] | 5.87[a] | 5.64[ab] | 6.50[bc] | 5.58[ab] | 5.75[ab] | 5.72[ab] | 6.71[c] | 5.23[a] | 5.98[abc] | *** | ns |
| **EC.S (uS/cm)** | 85.75[a] | 113.75[a] | 60.13[a] | 75.50[a] | 48.50[a] | 74.00[ab] | 46.50[a] | 48.50[a] | 123.00[bc] | 153.50[c] | 73.75[ab] | 102.50[abc] | ns | ns |
| **OC (%)** | 2.29[a] | 2.52[a] | 2.29[a] | 2.34[a] | 2.60[cd] | 2.89[d] | 2.78[d] | 2.51[bcd] | 1.97[ab] | 2.16[abc] | 1.79[a] | 2.16[abc] | ns | ns |
| **N (%)** | 0.19[a] | 0.205[a] | 0.185[a] | 0.196[a] | 0.208[cde] | 0.223[e] | 0.203[bcde] | 0.215[de] | 0.173[ab] | 0.188[abcd] | 0.168[a] | 0.178[abc] | ns | ns |
| **S (ppm)** | 16.37[a] | 8.00[a] | 15.59[a] | 14.04[a] | 10.09[ab] | 1.22[a] | 9.80[ab] | 8.10[ab] | 22.65[b] | 14.78[ab] | 21.39[b] | 19.97[b] | ns | ns |
| **P (ppm)** | 30.80[ab] | 42.31[b] | 16.97[a] | 20.18[a] | 35.75[a] | 39.08[a] | 14.55[a] | 19.23[a] | 25.86[a] | 45.55[a] | 19.38[a] | 21.14[a] | ** | ns |
| **K (ppm)** | 472.63[a] | 1077.25[b] | 453.13[a] | 541.63[a] | 339.00[a] | 994.25[bc] | 334.75[a] | 366.00[a] | 606.25[ab] | 1160.25[c] | 571.50[a] | 717.25[ab] | *** | ns |
| **Ca (ppm)** | 1462[a] | 2086[b] | 1438[a] | 1539[a] | 1765[ab] | 2315[b] | 1598[ab] | 1695[ab] | 1159[a] | 1858[ab] | 1279[a] | 1384[a] | ** | ns |
| **Mg (ppm)** | 248[a] | 342[b] | 260[a] | 245[a] | 250[ab] | 344[c] | 237[a] | 235[a] | 246[a] | 340[bc] | 283[abc] | 256[abc] | *** | ns |
| **Na (ppm)** | 21.63[a] | 32.73[a] | 18.03[a] | 18.34[a] | 7.17[ab] | 9.29[ab] | 4.48[a] | 5.70[ab] | 36.10[bc] | 56.18[c] | 31.58[abc] | 30.98[abc] | ns | ns |
| **Exch. Al (meq/100g)** | 0.07[a] | 0.04[a] | 0.19[a] | 0.11[a] | 0.78[ab] | 0.12[a] | 0.53[ab] | 0.04[a] | 0.06[ab] | 0.07[ab] | 0.33[b] | 0.17[ab] | ns | ns |
| **B (ppm)** | 0.58[a] | 0.96[b] | 0.55[a] | 0.68[a] | 0.54[a] | 0.93[ab] | 0.53[a] | 0.58[a] | 0.63[ab] | 0.99[b] | 0.58[a] | 0.78[ab] | *** | ns |
| **Mn (ppm)** | 434[a] | 443[a] | 446[a] | 429[a] | 567.50[b] | 533.50[b] | 575.75[b] | 553.75[b] | 300.50[a] | 353.25[a] | 315.25[a] | 303.75[a] | ns | * |
| **Fe (ppm)** | 89.25[b] | 70.19[a] | 83.70[a] | 77.33[ab] | 97.93[c] | 72.76[ab] | 89.63[bc] | 83.78[abc] | 80.58[ab] | 67.60[a] | 77.75[ab] | 70.88[a] | ** | ns |
| **Zn (ppm)** | 8.89[a] | 10.51[a] | 7.19[a] | 8.06[a] | 12.23[de] | 12.80[e] | 9.55[cd] | 10.80[cde] | 5.49[ab] | 8.23[bc] | 4.82[a] | 5.32[ab] | ns | ns |
| **Small Macro-aggregate (g)** | 48.11[ab] | 52.15[b] | 42.17[a] | 42.28[a] | 46.09[b] | 48.56[bc] | 36.53[a] | 36.76[a] | 50.15[bc] | 55.75[c] | 47.82[bc] | 47.80[bc] | ** | ns |
| **Micro-aggregate (g)** | 21.15[ab] | 17.43[a] | 28.66[b] | 27.13[b] | 25.58[bc] | 22.29[b] | 34.22[c] | 33.81[c] | 16.72[ab] | 12.58[a] | 23.10[b] | 20.46[ab] | * | ns |

Letters *a-d* designate significant differences at P ≤ 0.05. b) Means followed by the same letter are not significantly different. ns = not significant

*P ≤ 0.05

** P ≤ 0.01 and

*** P ≤ 0.001.

structure and presumptively function. The number of OTUs and alpha diversity analysis show with confidence that we achieved good coverage of the resident microbial diversity. Abundance of phylotypes affiliated to *Acidobacteria*, *Bacteroidetes*, *Chloroflexi*, *Cyanobacteria*, *Deinococcus-Thermus*, *Firmicutes*, *Fusobacteria*, *Gemmatimonadetes*, *Planctomycetes* and *Verrucomicrobia* were observed in this study. Members of these phyla are major contributors to soil biogeochemical processes and have also been reported in other studies [45]. Here, the authors describe the taxonomic composition of microbial community established in soil following long-term exposure to conventional and organic farming systems. Within the soil ecosystem, different groups perform varied functions hence a shift in the diversity and abundance due to effect of inputs on soil and plant health. Major families within *Proteobacteria* comprised *Rhodospirillaceae*, *Beijerinckiaceae*, *Burkholderiaceae* and *Bradyrhizobiaceae*. Some representatives of these families (e.g. *Burkholderiaceae*) are known to degrade recalcitrant organic matter in soil while other groups (e.g. *Beijerinckiaceae*) fix atmospheric nitrogen in the soil [46, 47]. At high relative abundance, these microbial groups could increase available nitrogen in organic farming system without fertilizer supplementation. *Actinobacteria* have been found to play a major role in organic matter turnover and carbon cycling. They can decompose recalcitrant carbon sources like cellulose and chitin and degrade herbicides and pesticides [48, 47]. In this study, Prokaryotic community composition and diversity analysis within sites and farming

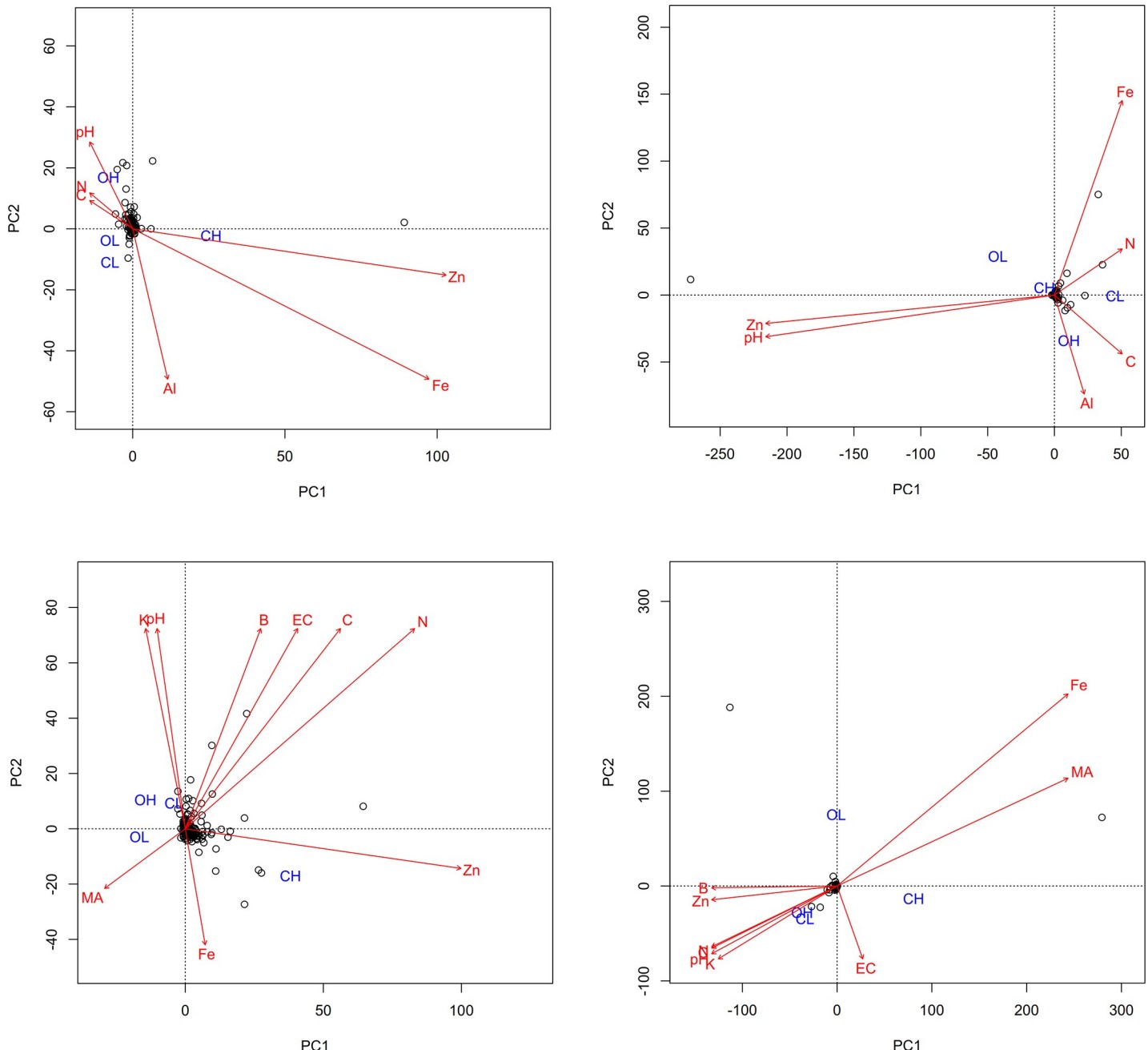

**Fig 6. a-d.** Principal component analysis of soil physicochemical characteristics that drive diversity within farming systems. OH, CH, OL and CL represents Org-High, Conv-High, Org-Low and Conv-Low farming systems.

systems displayed Thika site to harbor more shared and unique OTUs compared to Chuka site. This is a factor we attribute to soil aggregate composition and mineralogy. In both sites, conventional farming systems supported higher species richness although, there was no observable significant difference. This was attributed to integration of farmyard manure and inorganic fertilizer into the systems, promoting copiotropic prokaryotic groups to thrive due to high nutrient availability within the cropping season. On the other hand, low nutrient levels at the end of cropping season enhanced high abundance of unique prokaryotic groups

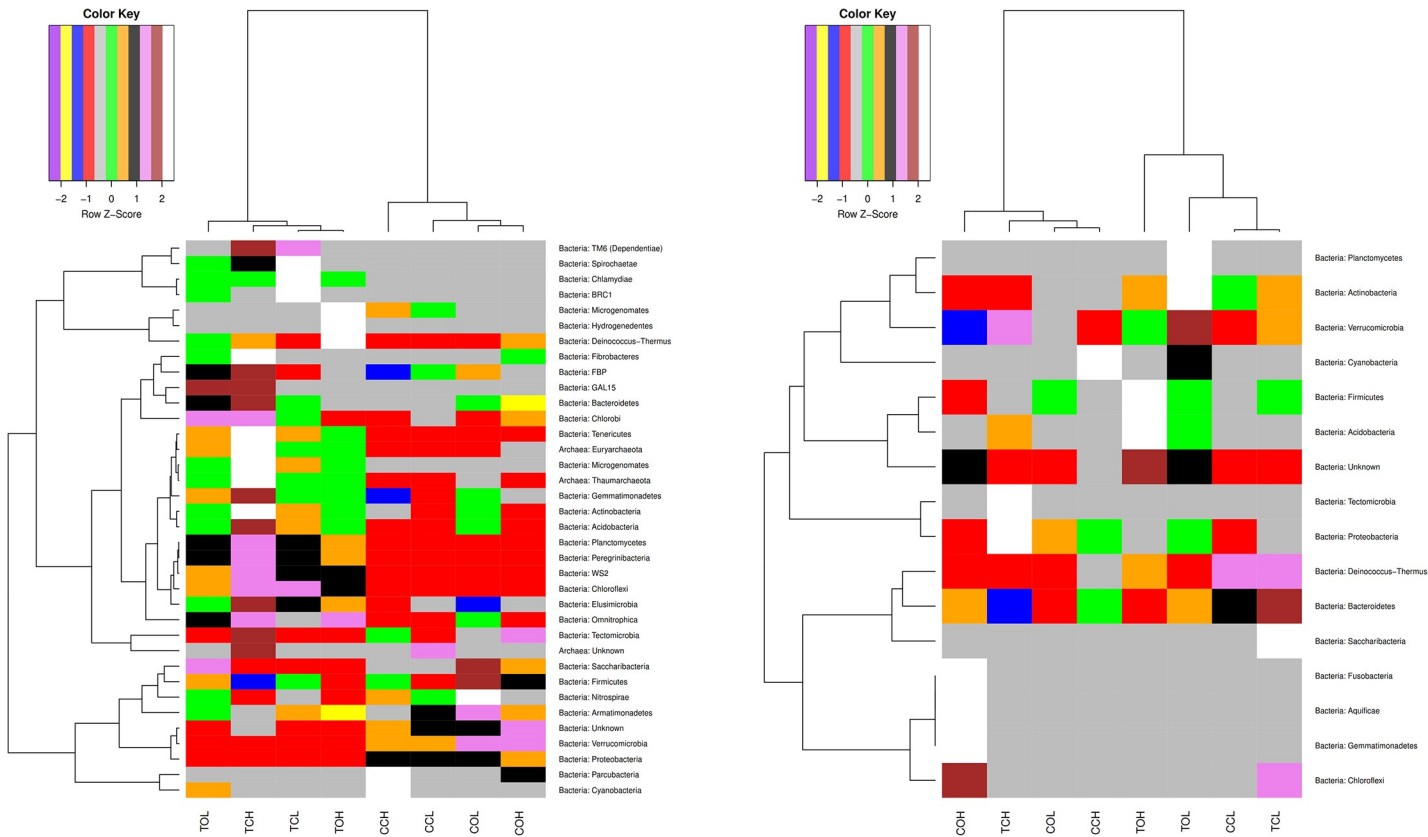

**Fig 7. a** and **b.** Hierarchical clustering of the most predominant prokaryotic taxa at phylum level within each farming system of 16S rDNA and 16S rRNA cDNA datasets in both sites. X-axis indicates the replicates within each system while the Y-axis indicates the taxonomic relationships. Total and active prokaryotic diversity is represented by **a** and **b,** respectively. CCL = Chuka Conv-Low; COL = Chuka Org-Low; CCH = Chuka Conv-High; COH = Chuka Org-High and TCL = Thika Conv-Low; TOL = Thika Org-Low; TCH = Thika Conv-High; TOH = Thika Org-High.

observed in conventional systems. Analysis of the 16S rRNA cDNA gives an indication of active microbial diversity at the time of sampling which explains the low OTU numbers in both sites (Chuka—390 and Thika—501 OTUs) as compared to 16S rDNA dataset. These could have been the communities carrying out the various biological processes within farming systems at the time. The low number of OTUs affiliated to active microbial diversity was attributed to lack of cropping activities within farming systems at the time of sampling. In this dataset, the most abundant phylotypes were affiliated to the classes *Alphaproteobacteria*, *Actinobacteria*, *Gammaproteobacteria*, *Betaproteobacteria*, *Acidimicrobia*, *Bacilli* and *Unknown bacterial phyla*. The unknown groups could form the basis for further studies in order to reveal their role within the farming systems.

Soil microbial activity has been reported to affect soil carbon dynamics by releasing carbon in form of carbon dioxide back into the atmosphere through respiration and is responsible for about 80–95% of carbon mineralization [49]. The presence of a higher number of unique OTUs and low organic carbon levels at Thika site as compared to Chuka site may be an indicator that higher species richness may eventually lead to carbon depletion through increased metabolic activities. Furthermore, Thika soils were found to contain higher sand content, a property that exposes soil organic carbon to heightened microbial activity [50]. The high amounts of organic carbon detected in the samples from Chuka confirms the findings of a previous study that indicated the soils found in humid regions contain more organic carbon than

soils within drier regions [51]. After six (6) years of continuous cropping within the trial sites, (Adamtey et al. unpublished results) pointed towards organic carbon build-up at Chuka and organic carbon depletion at Thika sites.

Clay minerals and oxides of Fe and Al have been shown to play important roles in adsorbing dissolved organic carbon [52, 53]. Since Thika soils contained high Fe levels coupled with high primary clay minerals, this may have created a stable environment for microbes to thrive. Chuka soils have been reported to contain the highest phyllosilicate clay minerals, especially kaolinite, involved in dissolved organic carbon preservation [54], making it unavailable for microbial attack and hence its build up at the site. In some occurrences within the current study, low input systems were found to harbor more OTUs than high input systems. This could be due to differences in soil macro-aggregates (> 250–2000 μM) and micro-aggregates (< 53–250 μM) (Adamtey et al. unpublished results). The high macro-aggregates may have provided unique environmental partitioning for soil microbiome which was isolated from its surroundings. Macro-aggregates are considered as massively concurrent incubators that allow enclosed microbial communities to pursue their own independent progression [55], hence creating more unique habitats for microbial colonization within these farming systems. Organic inputs not only carry various types of organic compounds, but also indigenous prokaryotes that remain in soil for a certain period of time [10]. Besides, incorporation of *Tithonia diversifolia* leaves and leaf extracts as well as *Lantana camara* leaves during composting and as starter N in organic farming systems could have lowered microbial diversity. These plants have been shown to contain anti-microbial properties resulting from steroids, saponins, tannins, polyphones and alkaloids which might be responsible for broad anti-bacterial activity [56, 57]. A significant prokaryotic community structuring based on farming systems was observed, probably reflecting variations in agricultural input amounts and management practices. This observation suggests a high degree of agro ecosystem microbiomic endemism and implies that each farming system harbors some degree of unique soil prokaryotic genetic resource. This result has significance in maximizing microbial functions in agroecosystems which has become a promising approach for the future of global agriculture. The data creates a better understanding in application of the benefits of soil microorganisms for resource uptake, plant growth, development and health, on agricultural production systems.

## Conclusion

This study revealed that farming systems have a profound impact on soil prokaryotic communities. Conventional farming systems were shown to support diverse prokaryotic communities compared to organic farming systems. It was also evident that prokaryotic diversity within the farming systems was influenced by complex interactions between a wide range of soil properties and agricultural inputs, demonstrating that prokaryotes within the soils are remarkably diverse. These inputs amend soil properties and microbial diversity, which in turn manipulates nutrient cycling processes altering soil fertility, plant productivity and environmental sustainability. Future studies should endeavor to build knowledge on soil and plant microbial biodiversity. This is in relation to common agronomic practices in different crop growth stages within farming systems, unravelling functional relations of soil-plant microbe interactions as well as developing strategies and tools for sustainable soil/plant management.

The aim for the future agricultural practices will be to safeguard agro-biodiversity by applying microbiome science in order to improve plant health, productivity, nutrient availability, and defense to diseases; and provide clear agricultural practices that will harness plant microbiomes for a sustainable agriculture and environment.

## Supporting information

**S1 Table. SysCom trials soil fertility management plan.**
(XLS)

**S2 Table. Chuka taxonomic order level.**
(XLSX)

**S3 Table. Thika taxonomic order level.**
(XLSX)

**S1 Data.**
(XLS)

## Acknowledgments

We acknowledge the support received from field assistants; Jane Makena and Felistus Mutua during field work.

## Author Contributions

**Conceptualization:** Edward Nderitu Karanja, Andreas Fliessbach, Noah Adamtey, Anne Kelly Kambura, Martha Musyoka, Komi Fiaboe, Romano Mwirichia.

**Data curation:** Edward Nderitu Karanja, Andreas Fliessbach, Anne Kelly Kambura, Romano Mwirichia.

**Formal analysis:** Edward Nderitu Karanja, Andreas Fliessbach, Noah Adamtey, Anne Kelly Kambura, Martha Musyoka, Komi Fiaboe, Romano Mwirichia.

**Funding acquisition:** Edward Nderitu Karanja, Andreas Fliessbach, Noah Adamtey, Komi Fiaboe.

**Investigation:** Edward Nderitu Karanja, Andreas Fliessbach, Noah Adamtey, Anne Kelly Kambura, Komi Fiaboe, Romano Mwirichia.

**Methodology:** Edward Nderitu Karanja, Andreas Fliessbach, Noah Adamtey, Anne Kelly Kambura, Martha Musyoka, Komi Fiaboe, Romano Mwirichia.

**Project administration:** Edward Nderitu Karanja, Noah Adamtey, Martha Musyoka, Komi Fiaboe.

**Resources:** Edward Nderitu Karanja, Noah Adamtey, Komi Fiaboe.

**Software:** Edward Nderitu Karanja, Anne Kelly Kambura, Romano Mwirichia.

**Supervision:** Andreas Fliessbach, Romano Mwirichia.

**Validation:** Edward Nderitu Karanja, Andreas Fliessbach, Noah Adamtey, Anne Kelly Kambura, Martha Musyoka, Komi Fiaboe, Romano Mwirichia.

**Visualization:** Edward Nderitu Karanja, Noah Adamtey, Anne Kelly Kambura, Martha Musyoka, Romano Mwirichia.

**Writing – original draft:** Edward Nderitu Karanja, Andreas Fliessbach, Noah Adamtey, Anne Kelly Kambura, Komi Fiaboe, Romano Mwirichia.

**Writing – review & editing:** Edward Nderitu Karanja, Andreas Fliessbach, Noah Adamtey, Anne Kelly Kambura, Martha Musyoka, Komi Fiaboe, Romano Mwirichia.

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
