## [Decision Letter · Decision Letter 0]

6 Apr 2020

PONE-D-19-32718

Diversity and structure of prokaryotic communities within organic and conventional farming systems in central highlands of Kenya

PLOS ONE

Dear Dr Mwirichia,

Thank you for submitting your manuscript to PLOS ONE. After careful consideration, we feel that it has merit but does not fully meet PLOS ONE’s publication criteria as it currently stands. Therefore, we invite you to submit a revised version of the manuscript that addresses the points raised during the review process.

The reviewer suggests some further analyses, a figure, such as a Venn Diagram, to visualize differences, and some improvements to the discussion. Those improvements will raise the manuscript to the technical standards in the field.

We would appreciate receiving your revised manuscript by May 21 2020 11:59PM. To enhance the reproducibility of your results, we recommend that if applicable you deposit your laboratory protocols in protocols.io, where a protocol can be assigned its own identifier (DOI) such that it can be cited independently in the future. For instructions see: http://journals.plos.org/plosone/s/submission-guidelines#loc-laboratory-protocols

We look forward to receiving your revised manuscript.

Kind regards,

Gabriel Moreno-Hagelsieb

Academic Editor

PLOS ONE

Journal Requirements:

2. We note that Figure 1  in your submission contain [map/satellite] images which may be copyrighted. All PLOS content is published under the Creative Commons Attribution License (CC BY 4.0), which means that the manuscript, images, and Supporting Information files will be freely available online, and any third party is permitted to access, download, copy, distribute, and use these materials in any way, even commercially, with proper attribution. For these reasons, we cannot publish previously copyrighted maps or satellite images created using proprietary data, such as Google software (Google Maps, Street View, and Earth). For more information, see our copyright guidelines: http://journals.plos.org/plosone/s/licenses-and-copyright.

a)    You may seek permission from the original copyright holder of Figure 1 to publish the content specifically under the CC BY 4.0 license.  

Reviewers' comments:

Reviewer's Responses to Questions

**Comments to the Author**

1. Is the manuscript technically sound, and do the data support the conclusions?

Reviewer #1: Partly

2. Has the statistical analysis been performed appropriately and rigorously? 

Reviewer #1: Yes

3. Have the authors made all data underlying the findings in their manuscript fully available?

Reviewer #1: Yes

4. Is the manuscript presented in an intelligible fashion and written in standard English?

Reviewer #1: No

5. Review Comments to the Author

Reviewer #1: Review

The analysis of microbial diversity inhabiting the soil and in direct relation to the plants is an interesting approach to evaluate under two different farming systems. The authors explain their samples come from a long-term exposition under different fertilization practices, and at different nutrient input levels (low and high). However, different crop rotation is seen throughout the years, where the total amounts of nutrients used for each treatment also varies with time and season.

Though the two different sites have similar geographical characteristics, such as altitude, temperature and cropping periods, soil type varies as well as the crops grown there. Soil type and plant species are known as the most important factors in shaping bacterial communities. Thus, the structure of microbial community in each site is thought to be different even before the experiment begins. Most of the analysis and discussion is based in trying to find similarities or differences in the diversity of microbial communities between the two sites, but because of the reasons above, interesting comparisons would better come from Conventional versus Organic and the different nutrient levels, regardless the site. Thereby, differences in microbial diversity, would be meaningful as the discussion will be strictly focused in the functioning of each community structure and the effects of the farming system in the microbial diversity. Also having an unfertilized plot would be a perfect control to compare both, physicochemical parameters and microbial diversity.

If comparing organic versus conventional microbial diversity per site, a Venn diagram would be a good alternative, showing the numbers at any taxa level (maybe genus) would help to discuss shared and unique ones.

During the discussion, I can highlight a few more considerations/questions for this study:

- High-throughput sequencing involved both 16S rDNA and 16S rRNA cDNA, but no more relevant differences in results or discussion is highlighted.

- What about the major differences of bacterial groups found in organic versus conventional farming? Were there some only found in one system but not in other worth to mention?

- Only organic microbial diversity is discussed.

- Most of the discussion is focused on differences of number of OTUs, but not in relevant community structure for each type of agricultural management. Thus, differences in chemical properties are attributed to the inherent and particular soil composition.

The authors conclude that Conventional systems harbor a higher diversity than Organics in both sites. However, similar studies have found that agricultural practices such as fertilization and tillage reduce the diversity of soil organisms, and some found that microbial diversity increases in organic farms (Tsiafouliet al.2015; Zhang et al., 2019). Therefore, combining synthetic with natural products provoked the uncertainty in the results. I.e. Conv-High and Org-low showed similar richness and Shannon index values in Tika (16S rDNA), Conv-low and Org-High had the same diversity (6.09) as shown in the Shannon index (Table 3). Differences in unique OTUs differed in Chuka site, where Conventional levels had more than Organic; however, the diversity index was again very similar. An additional Anova test would help to clarify those differences among farming systems.

Finding the unique OTUs could be key step in the analysis. The authors explained after filtering and trimming the sequences, they were clustered at 97%. This conventional step could have better results if eliminating exactly same sequences (dereplication) and counting all the remaining ones as true OTUs, to avoid underestimate some species.

Finally, the conclusion (line 327) is that conventional farming support (include more) diverse prokaryotic communities compared to organic farming systems. It is important to discuss if having more or less diversity would be more or less favourable in terms of farming sustainability since very recently studies support that having a more even community play an important role in soil ecosystem health, compared to richness.

6. PLOS authors have the option to publish the peer review history of their article (what does this mean?). If published, this will include your full peer review and any attached files.

Reviewer #1: No

---

## [Author Response · Author response to Decision Letter 0]

21 May 2020

Academic Editor’s Comment 1: Response from Authors: The authors would not like to make any changes to their financial disclosure.

Academic Editor’s Comment 2: To enhance the reproducibility of your results, we recommend that if applicable you deposit your laboratory protocols in protocols.io, where a protocol can be assigned its own identifier (DOI) such that it can be cited independently in the future. For instructions see: http://journals.plos.org/plosone/s/submission-guidelines#loc-laboratory-protocols

Response from Authors: The study methodology used standard protocols as referenced within the manuscript. Therefore, the authors do not wish to deposit any protocol.

Journal Requirements:

2. We note that Figure 1 in your submission contain [map/satellite] images which may be copyrighted. All PLOS content is published under the Creative Commons Attribution License (CC BY 4.0), which means that the manuscript, images, and Supporting Information files will be freely available online, and any third party is permitted to access, download, copy, distribute, and use these materials in any way, even commercially, with proper attribution. For these reasons, we cannot publish previously copyrighted maps or satellite images created using proprietary data, such as Google software (Google Maps, Street View, and Earth). For more information, see our copyright guidelines: http://journals.plos.org/plosone/s/licenses-and-copyright.

 a) You may seek permission from the original copyright holder of Figure 1 to publish the content specifically under the CC BY 4.0 license.

The following resources for replacing copyrighted map figures may be helpful: USGS National Map Viewer (public domain): http://viewer.nationalmap.gov/viewer/

Response from Authors: Figure 1 has been removed from the manuscript on lines 84 – 87 since the sites are also indicated by the GPS coordinates. 

Reviewer #1: Review

Reviewer Comment 1: The analysis of microbial diversity inhabiting the soil and in direct relation to the plants is an interesting approach to evaluate under two different farming systems. The authors explain their samples come from a long-term exposition under different fertilization practices, and at different nutrient input levels (low and high). However, different crop rotation is seen throughout the years, where the total amounts of nutrients used for each treatment also varies with time and season.

Response from Authors: OK

Reviewer Comment 2: Though the two different sites have similar geographical characteristics, such as altitude, temperature and cropping periods, soil type varies as well as the crops grown there. 

Response from Authors: Crops grown in both sites are the same.

Reviewer Comment 3: Soil type and plant species are known as the most important factors in shaping bacterial communities. Thus, the structure of microbial community in each site is thought to be different even before the experiment begins. Most of the analysis and discussion is based in trying to find similarities or differences in the diversity of microbial communities between the two sites, but because of the reasons above, interesting comparisons would better come from Conventional versus Organic and the different nutrient levels, regardless the site. Thereby, differences in microbial diversity, would be meaningful as the discussion will be strictly focused in the functioning of each community structure and the effects of the farming system in the microbial diversity. 

Response from Authors: In this study the comparison has been done at the following levels; (i) between farming systems (Organic versus Conventional) and the different nutrient levels, regardless of the site, (ii) at different input levels (high and low input levels) within and between sites. The sources of variation were System, Site and the interaction between System and Site. However, the diversity indicators within farming systems and sites showed no significant difference P>0.05. This is indicated within the manuscript in lines 223-224.It is important to note that high input farming systems were compared together while low input farming systems were compared together 

Reviewer Comment 4: Also having an unfertilized plot would be a perfect control to compare both, physicochemical parameters and microbial diversity.

Response from Authors: The Long Term Farming System Experiment was designed as a system comparison trial within and between sites for a period of 20 years. There is no unfertilized plot since it was not considered as a farming system.

Reviewer Comment 5: If comparing organic versus conventional microbial diversity per site, a Venn diagram would be a good alternative, showing the numbers at any taxa level (maybe genus) would help to discuss shared and unique ones.

Response from Authors: Further analyses have been carried out at order taxa level. A Venn diagram for each data set in both sites has been added to the manuscript in Fig 3a, 3g, 4a and 4g. The shared and unique taxa at order level within each farming system have also been shown on the pie charts (Fig. 3b - f, 3h – l, 4b – f and 4h - k) in lines 198 – 221 within the manuscript.

Reviewer Comment 6: During the discussion, I can highlight a few more considerations/questions for this study: 

High-throughput sequencing involved both 16S rDNA and 16S rRNA cDNA, but no more relevant differences in results or discussion is highlighted. What about the major differences of bacterial groups found in organic versus conventional farming? Were there some only found in one system but not in other worth to mention? Only organic microbial diversity is discussed.

Response from Authors: The results have been updated to show shared and unique prokaryotic taxa within 16S rDNA and 16S rRNA cDNA datasets in farming systems (lines 198 – 221). The discussion has also been enriched to cover both farming systems (line 292 – 323).

Reviewer Comment 7: Most of the discussion is focused on differences of number of OTUs, but not in relevant community structure for each type of agricultural management. Thus, differences in chemical properties are attributed to the inherent and particular soil composition.

Response from Authors: The community structure for each type of agricultural management system has been updated in the results section (lines 198 – 221). The discussion has also been enriched to cover the relevance of community structure in farming systems lines 285 – 319 and lines 351 – 359.

Reviewer Comment 8: The authors conclude that Conventional systems harbor a higher diversity than Organics in both sites. However, similar studies have found that agricultural practices such as fertilization and tillage reduce the diversity of soil organisms, and some found that microbial diversity increases in organic farms (Tsiafouli et al.2015; Zhang et al., 2019). Therefore, combining synthetic with natural products provoked the uncertainty in the results. I.e. Conv-High and Org-low showed similar richness and Shannon index values in Tika (16S rDNA), Conv-low and Org-High had the same diversity (6.09) as shown in the Shannon index (Table 3). Differences in unique OTUs differed in Chuka site, where Conventional levels had more than Organic; however, the diversity index was again very similar. An additional Anova test would help to clarify those differences among farming systems. 

Response from Authors: 

Combining synthetic inputs with natural products within conventional systems possibly aggravated the uncertainty in the results, i.e. Conv-High and Org-low showed similar richness and Shannon index values in Thika (16S rDNA), Conv-low and Org-High had the same diversity (6.09) as shown in the Shannon index (Table 3). Although there were OTU differences among farming systems as shown in Table 3, alpha diversity indices within farming systems and sites showed no significant difference (P>0.05) in Richness (S) and Shannon index (H’); (Lines 223-224). Similarly, there were no significant differences observed at Thika site (ANOSIM P<0.672 and 0.241 within 16S rDNA and 16S rRNA cDNA datasets, respectively); Lines 232-234. However, Analysis of Similarity pointed to highly significant differences between OTUs within high and low input farming systems (P<0.001) at Chuka site Lines 230-232.

Reviewer Comment 9: Finding the unique OTUs could be key step in the analysis. The authors explained after filtering and trimming the sequences, they were clustered at 97%. This conventional step could have better results if eliminating exactly same sequences (dereplication) and counting all the remaining ones as true OTUs, to avoid underestimate some species.

Response from Authors: The FASTQ sequences were demultiplexed and quality checked using a pipeline that denoises sequences, removes chimeras, picks representative sequences, calculates denoising statistics and creates an OTU table. During this step, sequences which were <200 base pairs after phred20- base quality trimming, sequences with ambiguous base calls, and those with homopolymer runs exceeding 6bp were removed. At this step, any exact same sequences (dereplication) were removed. The denoising statistics indicated satisfactory level of sequence quality and there was adequate representation of all high-quality sequences within the samples. 

Reviewer Comment 10:

Finally, the conclusion (line 327) is that conventional farming support (include more) diverse prokaryotic communities compared to organic farming systems. It is important to discuss if having more or less diversity would be more or less favourable in terms of farming sustainability since very recently studies support that having a more even community play an important role in soil ecosystem health, compared to richness.

Response from Authors: The conclusion has been enriched appropriately in lines 361-376.

Academic Editor’s Comment 2: While revising your submission, please upload your figure files to the Preflight Analysis and Conversion Engine (PACE) digital diagnostic tool, https://pacev2.apexcovantage.com/. PACE helps ensure that figures meet PLOS requirements. To use PACE, you must first register as a user. Registration is free. Then, login and navigate to the UPLOAD tab, where you will find detailed instructions on how to use the tool. If you encounter any issues or have any questions when using PACE, please email us at figures@plos.org. Please note that Supporting Information files do not need this step.

Response from Authors: All figures have been uploaded to the Preflight Analysis and Conversion Engine (PACE) digital diagnostic tool and the corrected figures have been filed for use in this manuscript. However, there were no PACE adjustments on pie diagrams uploaded on PACE.

We believe to have adequately responded on all the comments raised regarding this manuscript in readiness for its publication to your journal. 

---

## [Decision Letter · Decision Letter 1]

12 Jun 2020

PONE-D-19-32718R1

Diversity and structure of prokaryotic communities within organic and conventional farming systems in central highlands of Kenya

PLOS ONE

Dear Dr. Mwirichia,

Thank you for submitting your manuscript to PLOS ONE. After careful consideration, we feel that it has merit but does not fully meet PLOS ONE’s publication criteria as it currently stands. Therefore, we invite you to submit a revised version of the manuscript that addresses the points raised during the review process.

While the main problems with the oroginal manuscript were addressed, the reviewer found a few inconsistencies, for example, between what the data indicate and the conclusions. Please address all of these inconsistencies.

We look forward to receiving your revised manuscript.

Kind regards,

Gabriel Moreno-Hagelsieb

Academic Editor

PLOS ONE

Reviewers' comments:

Reviewer's Responses to Questions

**Comments to the Author**

1. If the authors have adequately addressed your comments raised in a previous round of review and you feel that this manuscript is now acceptable for publication, you may indicate that here to bypass the “Comments to the Author” section, enter your conflict of interest statement in the “Confidential to Editor” section, and submit your "Accept" recommendation.

Reviewer #1: All comments have been addressed

2. Is the manuscript technically sound, and do the data support the conclusions?

Reviewer #1: Yes

3. Has the statistical analysis been performed appropriately and rigorously? 

Reviewer #1: No

4. Have the authors made all data underlying the findings in their manuscript fully available?

Reviewer #1: No

5. Is the manuscript presented in an intelligible fashion and written in standard English?

Reviewer #1: Yes

6. Review Comments to the Author

Reviewer #1: In your discussion you clearly express that conventional farming systems supported significantly higher species richness in both sites, which in fact was not supported by any of the index, meaning no significant difference in Richness and Shannon index. The only difference was seen between OTUs within high and low input farming systems at Chuka.

Information provided for Thika taxonomical orders differ between the results. In Fig 3, 213 taxa (Order) are supposed to be shown, but the Venn diagram contains 229. Furthermore, only 62 orders are shown in S3 Table, which corresponds to the list provided for this site. On the other hand, there is an accurate correspondence of these information for Chuka site.

What is the order of pie charts? It is not mention in the figure legend.

Please review:

Line 57: hence,

Line 312: to16S

Line 294: the idea is not clear, maybe substitute effect for affect.

7. PLOS authors have the option to publish the peer review history of their article (what does this mean?). If published, this will include your full peer review and any attached files.

Reviewer #1: No

---

## [Author Response · Author response to Decision Letter 1]

28 Jun 2020

Reviewer #1:

Comments to the Author

Reviewer Comment 1: In your discussion you clearly express that conventional farming systems supported significantly higher species richness in both sites, which in fact was not supported by any of the index, meaning no significant difference in Richness and Shannon index. The only difference was seen between OTUs within high and low input farming systems at Chuka.

Response from Authors: The sentence has been rephrased to reflect the results (Lines 304-309).

Reviewer Comment 2: Information provided for Thika taxonomical orders differ between the results. In Fig 3, 213 taxa (Order) are supposed to be shown, but the Venn diagram contains 229. 

Response from Authors: Thika taxonomical orders are 229 as shown in the Venn diagram. This correction has been effected in line 179. We apologize for the typing error.

Reviewer Comment 3: Furthermore, only 62 orders are shown in S3 Table, which corresponds to the list provided for this site. On the other hand, there is an accurate correspondence of these information for Chuka site.

Response from Authors: During data analysis, all the samples that scored a minimum relative abundance of 5 % and above in at least one (1) sample per farming system were selected for diagram presentation. All samples scoring below 5 % across all farming systems were reported in the narrative in lines 170 - 184. Therefore, in supplementary table S3, sixty two (62) orders had scored at least 5% in one sample within the farming system. 

Reviewer Comment 4: What is the order of pie charts? It is not mention in the figure legend.

Response from Authors: The pie charts were made at order level. This has been added to the figure legend.

Reviewer Comment 5: Please review: Line 57: hence,

Response from Authors: A comma has been added after the word “hence” in Line 57.

Reviewer Comment 6: Please review: Line 312: to16S

Response from Authors: A space has been added to separate “to16S” in Line 315.

Reviewer Comment 7: Please review: Line 294: the idea is not clear, maybe substitute effect for affect.

Response from Authors: The sentence has been rephrased to make it clear.

Reviewer Comment 8. Have the authors made all data underlying the findings in their manuscript fully available? The PLOS Data policy requires authors to make all data underlying the findings described in their manuscript fully available without restriction, with rare exception (please refer to the Data Availability Statement in the manuscript PDF file). The data should be provided as part of the manuscript or its supporting information, or deposited to a public repository. For example, in addition to summary statistics, the data points behind means, medians and variance measures should be available. If there are restrictions on publicly sharing data—e.g. participant privacy or use of data from a third party—those must be specified.

Response from Authors: The sequences were submitted to NCBI Sequence Read Archive. The SRA accession numbers are as indicated in lines 146-150. Statistical data analysis summaries for soil physicochemical characteristics were as shown in Table 4. Analysis of variance (ANOVA) was performed at P < 0.05, 0.01 and 0.001 using a linear mixed-effect model with farming system and site as fixed effects, while replication was used as random effect. Least mean squares were computed and means separated using Tukey’s ad hoc method. Letters a-d were used to designate significant differences between means at P ≤ 0.05. The authors have availed the soil physicochemical raw data as an attachment for purposes of information in order to show the values behind means variance measures.

---

## [Editor Report · Decision Letter 2]

10 Jul 2020

Diversity and structure of prokaryotic communities within organic and conventional farming systems in central highlands of Kenya

PONE-D-19-32718R2

Dear Dr. Mwirichia,

We’re pleased to inform you that your manuscript has been judged scientifically suitable for publication and will be formally accepted for publication once it meets all outstanding technical requirements.

Kind regards,

Gabriel Moreno-Hagelsieb

Academic Editor

PLOS ONE
---

## [Editor Report · Acceptance letter]

14 Jul 2020

PONE-D-19-32718R2 

Diversity and structure of prokaryotic communities within organic and conventional farming systems in central highlands of Kenya 

Dear Dr. Mwirichia:

I'm pleased to inform you that your manuscript has been deemed suitable for publication in PLOS ONE. Congratulations! Your manuscript is now with our production department. 

Kind regards, 

on behalf of

Prof. Gabriel Moreno-Hagelsieb 

Academic Editor

PLOS ONE